# *Acinetobacter baumannii* under Acidic Conditions Induces Colistin Resistance through PmrAB Activation and Lipid A Modification

**DOI:** 10.3390/antibiotics12050813

**Published:** 2023-04-26

**Authors:** Seo-Yeon Ko, Nayeong Kim, Seong-Yong Park, Seong-Yeop Kim, Minsang Shin, Je-Chul Lee

**Affiliations:** Department of Microbiology, School of Medicine, Kyungpook National University, Daegu 41944, Republic of Korea; sygogo97@knu.ac.kr (S.-Y.K.); tbc02021@knu.ac.kr (N.K.); psyseongyong@knu.ac.kr (S.-Y.P.); dgsk0207@knu.ac.kr (S.-Y.K.); shinms@knu.ac.kr (M.S.)

**Keywords:** PmrAB, *Acinetobacter baumannii*, lipopolysaccharide, lipid A, colistin

## Abstract

Colistin is a last-resort antimicrobial agent for treating carbapenem-resistant *Acinetobacter baumannii* infections. The activation of PmrAB by several environmental signals induces colistin resistance in Gram-negative bacteria. This study investigated the molecular mechanisms of colistin resistance in *A. baumannii* under acidic conditions using wild-type (WT) *A. baumannii* 17978, Δ*pmrA* and Δ*pmrB* mutants, and *pmrA*-complemented strains. The *pmrA* or *pmrB* deletion did not affect the growth of *A. baumannii* under acidic or aerobic conditions. *A. baumannii* under acidic (pH 5.5) and high-iron (1 mM) conditions showed 32- and 8-fold increases in the minimum inhibitory concentrations (MICs) of colistin, respectively. The Δ*pmrA* and Δ*pmrB* mutants at pH 5.5 showed a significant decrease in colistin MICs compared to the WT strain at pH 5.5. No difference in colistin MICs was observed between WT and mutant strains under high-iron conditions. The *pmrCAB* expression significantly increased in the WT strain at pH 5.5 compared to the WT strain at pH 7.0. The *pmrC* expression significantly decreased in two mutant strains at pH 5.5 compared to the WT strain at pH 5.5. The PmrA protein was expressed in the Δ*pmrA* strain carrying ppmrA_FLAG plasmids at pH 5.5 but not at pH 7.0. Lipid A modification by the addition of phosphoethanolamine was observed in the WT strain at pH 5.5. In conclusion, this study demonstrated that *A. baumannii* under acidic conditions induces colistin resistance via the activation of *pmrCAB* operon and subsequent lipid A modification.

## 1. Introduction

*Acinetobacter baumannii* is a Gram-negative, non-fermenting pathogen that causes various nosocomial infections, especially in severely ill patients admitted to intensive care units. This microorganism is a low-virulent pathogen but poses a global threat to human health through antimicrobial resistance [1,2]. The World Health Organization declared carbapenem-resistant *A. baumannii* as the first priority pathogen for which the development of new antimicrobial agents is urgently needed [3]. Due to the prevalence of multidrug-resistant (MDR) and carbapenem-resistant *A. baumannii* strains, colistin is one of the last therapeutic options to treat MDR *A. baumannii* infections [4]. However, colistin-resistant *A. baumannii* has been increasingly reported worldwide [5,6]. The complete loss of lipopolysaccharide (LPS) by mutations in enzymes involved in the lipid A biosynthetic pathway (*lpxA*, *lpxC*, and *lpxD*) and LPS modification by mutations in *pmrAB* are the two main mechanisms of colistin resistance in *A. baumannii* [7,8,9].

Bacteria have several signaling mechanisms for adapting to various environmental conditions. The two-component system (TCS) is an important mediator of signal transduction by which bacteria sense and respond to environmental stimuli. TCS plays a central role in bacterial adaptation responses that control antimicrobial resistance, cell-to-cell communication, pathogenesis, and bacterial survival [10,11]. Genome analysis of clinical *A. baumannii* isolates revealed approximately 20 genes encoding TCS [12]. Of the well-characterized proteins of TCS in *A. baumannii*, PmrAB is associated with resistance to polymyxins by regulating genes involved with lipid A modification [13]. PmrAB consists of a sensor kinase, PmrB, and a response regulator, PmrA, encoded by the *pmrCAB* operon [14]. The *pmrAB* activation leads to the upregulation of *pmrC* (also called *eptA*), *naxD*, and *arnBCADTEF-pmrE* operon (also called *pmrHFIJKLM-ugd*), encoding phosphoethanolamine transferase, N-acetylgalactosamine deacetylase (GalNAc), and 4-amino-4-deoxy-L-arabinose (L-Ara4N) transferase, respectively [12,13,14]. PhoPQ TCS activates *pmrAB* indirectly through *pmrD*, but the PhoPQ and L-Ara4N biosynthetic pathways are absent in *A. baumannii* [14,15]. PmrAB is activated by several environmental signals such as acidic pH, high iron, or low magnesium levels, leading to the induction of resistance to polymyxins through lipid A modification by *pmrC* expression in Gram-negative bacteria, including *Salmonella enterica*, *Escherichia coli,* and *Klebsiella pneumoniae* [16,17,18,19]. Adams et al. [13] showed that colistin susceptibility decreased in *A. baumannii* under low pH or iron supplementation. However, little is known about the induction of colistin resistance associated with the *pmrCAB* operon in *A. baumannii* exposed to different environmental factors. As *A. baumannii* has to adapt to and survive in harsh environments during infection, it is important to understand the regulatory mechanisms of colistin resistance in *A. baumannii* exposed to various environmental conditions. This study investigated the molecular mechanisms of colistin resistance by activating the *pmrCAB* operon in *A. baumannii* under acidic conditions.

## 2. Results

### 2.1. Construction of ΔpmrA and ΔpmrB Mutants and pmrA-Complemented Strains

To investigate whether the induction of colistin resistance in *A. baumannii* exposed to different environmental conditions was mediated by PmrAB TCS, Δ*pmrA* (HJ2751D) and Δ*pmrB* (HJ2750D) mutant strains were constructed using the wild-type (WT) *A. baumannii* ATCC 17978 (Appendix A). The *pmrA*-complemented strain (HJ2751C) was constructed by introducing the pWH1266 plasmid carrying the *ompA* promoter, *pmrA* gene, and T1 terminator into the HJ2751D strain (Appendix A). The *ompA* promoter was inserted in the pWH1266 plasmid for high *pmrA* gene expression. The deletion and complementation of genes were confirmed by polymerase chain reaction (PCR) analysis. The expected amplicon sizes of 3444, 2775, 2115, and 1422 bp were observed in WT, HJ2751D, HJ2750D, and HJ2751C strains, respectively (Appendix A).

### 2.2. Effects of pmrAB on the Growth of A. baumannii

To investigate whether *pmrAB* could affect the growth of *A. baumannii*, bacterial growth was monitored in WT, HJ2751D, and HJ2750D strains under static or shaking conditions. Bacterial strains were grown in lysogeny broth (LB), where the initial pH of the culture medium was 7.0 and 5.5 separately. Here, pH 7.0 was selected as a control for acidic pH. No significant difference in bacterial growth was observed among WT, HJ2751D, and HJ2750D strains at pH 7.0 or 5.5 (Figure 1). These results suggest that PmrAB does not affect the growth of *A. baumannii* under acidic pH or aerobic conditions.

### 2.3. Induction of Colistin Resistance in A. baumannii under Acidic or High-Iron Conditions

To determine the effect of pH on the induction of colistin resistance in *A. baumannii*, four *A. baumannii* strains (WT, HJ2751D, HJ2750D, and HJ2751C) were grown in cation-adjusted Mueller–Hinton broth (CAMHB) of different pH levels (pH 7.0, 6.5, 5.5, and 4.5). The minimum inhibitory concentrations (MICs) of colistin were 1 μg/mL in WT, HJ2751D, and HJ2750D strains at pH 7.0 and 6.5, whereas the WT strain grown at pH 5.5 showed a dramatic increase in colistin MIC (32 μg/mL) (Table 1). However, colistin MICs in HJ2751D and HJ2750D at pH 5.5 increased only 2-fold compared to those in strains at pH 7.0. No growth was observed in *A. baumannii* strains grown at pH 4.5. The *pmrA* complementation in the HJ2751D strain restored the colistin MIC (256 μg/mL) at pH 5.5. Next, to evaluate the effects of iron on the induction of colistin resistance, *A. baumannii* strains were grown in CAMHB supplemented with 100 μM, 500 μM, and 1 mM ferric chloride. With increasing ferric chloride concentrations, colistin MICs of the WT strain at pH 7.0 increased from 1 to 8 μg/mL. However, colistin MICs were not different among WT, HJ2751D, HJ2750D, and HJ2751C strains at 500 μM or 1 mM ferric chloride. These results suggest that *A. baumannii* induces colistin resistance under acidic or high-iron conditions. PmrAB is responsible for the induction of colistin resistance in *A. baumannii* under acidic conditions, but colistin resistance in *A. baumannii* under high-iron conditions is independent of PmrAB.

### 2.4. High Expression of pmrCAB Operon and Production of PmrA in A. baumannii under Acidic Conditions

To investigate whether colistin resistance in *A. baumannii* under acidic conditions was directly mediated by PmrAB TCS, quantitative real-time PCR (qPCR) was performed to determine the expression of the *pmrCAB* operon in WT, HJ2751D, and HJ2750D strains grown at pH 7.0 and 5.5 individually. The WT strain at pH 5.5 exhibited a 2.6-fold increase in *pmrA* expression, a 2.1-fold increase in *pmrB*, and an 8.3-fold increase in *pmrC* compared to the WT strain at pH 7.0 (Figure 2). The *pmrA* and *pmrB* expressions increased in HJ2750D (Δ*pmrB* mutant) and HJ2751D (Δ*pmrA* mutant) strains at pH 7.0 or 5.5, respectively. The *pmrC* expression significantly increased in the HJ2751D strain at pH 7.0 compared to that in the WT strain at pH 7.0, but the colistin MICs of the two strains were the same (1 μg/mL). The *pmrA*, *pmrB*, and *pmrC* expressions in the HJ2750D and HJ2751D mutant strains at pH 5.5 significantly decreased compared to those in the WT strain at pH 5.5. Next, to determine whether the production of response regulator PmrA increased in *A. baumannii* under acidic conditions due to high *pmrA* expression, western blot analysis was performed in HJ2751D and HJ2750D strains carrying ppmrA_FLAG plasmids. The ppmrA_FLAG plasmids carried the C-terminal FLAG-tagged *pmrA* with its promoter in the pWH1266 vector (Figure 3A). PmrA-FLAG-tagged proteins were only detected in the HJ2751D strain carrying ppmrA_FLAG plasmids at pH 5.5 but not at pH 7.0 (Figure 3B). However, they were not detected in the HJ2750D strain carrying ppmrA_FLAG plasmids at pH 5.5 or 7.0. These results suggest that *pmrAB* is upregulated in *A. baumannii* under acidic conditions, which induces *pmrC* upregulation.

### 2.5. Lipid A Modification in A. baumannii under Acidic Conditions

To determine whether *A. baumannii* under acidic conditions modified lipid A of LPS, matrix-assisted laser desorption ionization-time of flight mass spectrometry (MALDI-TOF MS) was performed. Lipid A of LPS was extracted from the WT strain grown at pH 7.0 or 5.5. Four major ion peaks, *m/z* 1712, 1728, 1894, and 1910, were observed in lipid A extracted from the WT strain at pH 7.0 (Figure 4A). The lipid A structure at *m/z* 1728 is a species of hexa-acylated, bis-phosphorylated lipid A with two phosphates and two 2-amino-2-deoxyglucose residues with three 12:0 (3-OH), two 14:0 (3-OH), and one 12:0 [20]. The lipid A structure at *m/z* 1910 is a species of hepta-acylated, bis-phosphorylated lipid A with two phosphates and two 2-amino-2-deoxyglucose residues with three 12:0 (3-OH), two 14:0 (3-OH), and two 12:0. The ion peaks with *m/z* 1712 and 1894 are hexa- and hepta-acylated bis-phosphorylated lipid A, where the hydroxyl group is eliminated (*m/z* −16) with substitution of one 14:0 (3-OH) with one 14:0. Lipid A from the WT strain at pH 5.5 showed four additional ion peaks with *m/z* 1835, 1851, 2017, and 2033 (Figure 4B). These additional ion peaks represented a shift of *m/z* +123 from the four main peaks, corresponding to the addition of a single phosphoethanolamine residue [9]. However, lipid A from the HJ2750D and HJ2751D mutant strains showed the same ion peaks as the WT strain (Figure 4C,D). These results suggest that *A. baumannii* under acidic conditions modifies lipid A by the addition of phosphoethanolamine, resulting in the induction of colistin resistance.

## 3. Discussion

This study investigated colistin resistance linked to PmrAB TCS in *A. baumannii* in response to acidic or high-iron conditions. Colistin resistance in *A. baumannii* under acidic conditions was induced by the upregulation of the *pmrCAB* operon and subsequent lipid A modification by the addition of phosphoethanolamine. Colistin resistance was also induced in *A. baumannii* under high-iron conditions, but it was independent of PmrAB.

Colistin is a last-resort drug for treating carbapenem-resistant *A. baumannii* infections [4]. The antibacterial activity of colistin occurs through the displacement of divalent cations from the phosphate groups of LPS [21]. The electrostatic interaction between the positively charged colistin and negatively charged phosphate groups of LPS stabilizes the complex. The insertion of colistin into the outer membrane induces membrane disruption and the leakage of intracellular contents, leading to bacterial death [22]. Colistin resistance is primarily mediated by the modification of lipid A-head groups to reduce the electrostatic interaction of colistin with LPS. The addition of positively charged residues, such as L-Ara4N and phosphoethanolamine, to lipid A decreases the net negative charge of lipid A to 0 and −1.5 to −1 on the bacterial surface, respectively [23]. LPS modification by PhoPQ or PmrAB activation through mutations or environmental signals leads to the overexpression of genes associated with lipid A modification [24]. Beceiro et al. [25] demonstrated that the development of colistin resistance in *A. baumannii* requires point mutations in *pmrB*, upregulation of *pmrAB*, and expression of *pmrC*, leading to the addition of phosphoethanolamine to lipid A. In addition, PmrAB TCS is activated by various environmental signals such as acidic pH, high iron levels, and low magnesium concentrations in Gram-negative bacteria [26,27,28]. Low magnesium conditions in *A. baumannii* slightly upregulate *pmrA* expression and lipid A modification by the addition of ion peaks with *m/z* 2034 in MALDI-TOF MS [25], suggesting that PmrAB is activated by low magnesium conditions in *A. baumannii*.

In this study, ferric chloride supplementation increased the colistin MICs against the WT *A. baumannii* strain in a dose-dependent manner. No difference in the colistin MICs was observed between WT and Δ*pmrA* or Δ*pmrB* mutant strains, suggesting no association of colistin resistance with PmrAB in *A. baumannii* under high-iron conditions. However, *A. baumannii* grown at pH 5.5 increased the colistin MIC (32-fold) compared to bacteria grown at pH 7.0. This result was consistent with the previous study that showed an increase in the colistin MIC of the *A. baumannii* ATCC 17978 strain grown at pH 5.5 [13]. This study demonstrated that the WT strain at pH 5.5 exhibited *pmrB*, *pmrA*, and *pmrC* upregulation compared to the WT strain at pH 7.0. However, the previous study did not show a significant difference in *pmrA* expression between *A. baumannii* at pH 7.7 and 5.5, although *pmrA* expression in *A. baumannii* at pH 5.5 slightly increased compared to that at pH 7.7 [13]. The difference between the two studies was medium pH and bacterial number for RNA extraction; bacteria were grown to an optical density at 600 nm (OD_600_) of 0.25 and 0.5 in this study and the previous study, respectively. This study demonstrated the production of PmrA proteins in the Δ*pmrA* mutant carrying *pmrA* at pH 5.5, but they were not detected in the same bacteria grown at pH 7.0 or the Δ*pmrB* mutant carrying *pmrA* at pH 7.0 or 5.5. These results suggest that acidic conditions induce the production of a response regulator, PmrA, and that a sensor kinase, PmrB, is essential to express PmrA proteins. Furthermore, the WT strain at pH 5.5 showed a significant increase in the *pmrC* expression compared to the WT strain at pH 7.0. *A. baumannii* ATCC 17978 grown at pH 5.5 induced the enzymatic activity of phosphoethanolamine transferase to modify lipid A, but the addition of phosphoethanolamine to lipid A was not observed in the Δ*pmrA* or Δ*pmrB* mutant strains at pH 5.5. PmrAB TCS also regulates *naxD* [29]. NaxD deacetylates N-acetylgalactosamine to galactosamine. This conversion step is required for the addition of galactosamine to lipid A. However, lipid A modification by the addition of galactosamine (*m/z* 2071) was not observed in the WT strain grown at pH 5.5.

In conclusion, this study demonstrated that *A. baumannii* upregulates *pmrCAB* expression and modifies lipid A under acidic conditions by the addition of phosphoethanolamine, leading to colistin resistance. Colistin resistance is also induced by high iron levels, but this phenomenon is independent of PmrAB in *A. baumannii*. This study provides insights into the molecular mechanisms of colistin resistance in *A. baumannii* in harsh environmental conditions such as acidic pH. As PmrAB also regulates virulence determinants responsible for pathogenicity [30,31,32], the complex regulatory network of *A. baumannii* in response to different environmental factors should be further investigated.

## 4. Materials and Methods

### 4.1. Bacterial Strains, Plasmids, and Growth Media

The bacterial strains and plasmids used in this study are listed in Table 2. Bacteria were cultured in LB (Bioshop, Burlington, ON, Canada) or LB agar plates. Bacterial culture media were adjusted to a desired pH with HCl (Duksan Pure Chemicals, Ansan, Republic of Korea) or NaOH (Duksan Pure Chemicals). MHB (Difco, Detroit, MI, USA) was used to determine the susceptibility of colistin (Sigma-Aldrich, St. Louis, MO, USA). Kanamycin (50 μg/mL), chloramphenicol (20 μg/mL), or tetracycline (10 μg/mL) were added to the growth medium to maintain plasmids in bacteria and select the mutant or complementary colonies.

### 4.2. Construction of ΔpmrA and ΔpmrB Mutants and pmrA-Complemented Strains

Δ*A1S_2751* (*pmrA*) and Δ*A1S_2750* (*pmrB*) mutant strains were constructed using the WT *A. baumannii* ATCC 17978 by the markerless gene deletion method as previously described [33]. *pmrA* complementation in the HJ2751D strain was conducted using an overlap extension PCR. The specific PCR primers used to construct the mutant and complemented strains are listed in Appendix A. The construction of mutant and complemented strains is presented in Appendix A and Methods in detail.

### 4.3. Construction of ppmrA_FLAG Plasmids

The *pmrA* coding region with a FLAG epitope sequence just before the stop codon, its promoter, and T1 terminator was amplified using primer sets of EcoRI_pmrAB_promoter (F)/Promoter_2751 (R), Promoter_2751 (F)/2751_FLAG (R), and FLAG_T1 (F)/T1_PstI (R), respectively (Appendix A). Three PCR products were assembled by an overlap extension PCR using EcoRI_pmrAB_promoter (F)/T1_PstI (R) primers. The PCR fragments were digested with *EcoR*I and *Pst*I and introduced into the pWH1266 plasmids to construct ppmrA_FLAG plasmids.

### 4.4. Antimicrobial Susceptibility Testing

The MICs of colistin were determined by the broth microdilution method according to the guidelines of the Clinical Laboratory Standards Institute [34]. *E. coli* ATCC 25922 and *Pseudomonas aeruginosa* ATCC 27853 were used as quality control strains. Colistin susceptibility testing was performed in 96-well plates containing CAMHB supplemented with different concentrations of colistin. MICs were defined as the lowest colistin concentration that inhibits the visible growth of bacteria.

### 4.5. Bacterial Growth Curve

*A. baumannii* strains were grown in LB overnight at 37 °C and diluted to an OD_600_ of 1.0. The bacterial suspension was diluted to 1:20 in fresh LB, where the initial pH was 7.0 or 5.5, and cultured under shaking or static conditions for 48 h. Bacteria were sampled at 4 h intervals, and OD_600_ was measured using a spectrophotometer (Biochrom, Cambridge, UK). The bacterial growth assay was performed in two independent experiments.

### 4.6. Western Blot Analysis

The FLAG-tagged PmrA protein was detected by western blot analysis as previously described with some modifications [35]. Bacteria were grown in LB overnight at 37 °C. Cultured bacteria were inoculated into a fresh LB with pH 7.0 or 5.5, where the initial OD_600_ was 0.02, and cultured at 37 °C with shaking conditions. Bacteria were harvested at an OD_600_ of 0.25 and resuspended in a B-PER (Thermo Scientific, Waltham, MA, USA) containing an ethylenediaminetetraacetic acid-free protease inhibitor cocktail (GenDEPOT, Barker, TX, USA). After incubation for 15 min at room temperature, cell debris was removed by centrifugation. The protein concentration was determined using the Pierce™ BCA Protein Assay Kit (Thermo Scientific). Equal amounts of cell lysates were suspended in sodium dodecyl sulfate (SDS) sample buffer (250 mM Tris HCl (pH 6.8), 8% SDS, 0.02% bromophenol blue, glycerol, and β-mercaptoethanol), and the samples were run on a 10% polyacrylamide gel. The proteins were transferred onto a polyvinylidene fluoride membrane by electroblotting. The membrane was blocked with 5% skim milk in Tris-buffered saline with 0.05% Tween-20 (TBST) for 2 h. Membranes were incubated with a FLAG epitope-tagged (DYKDDDDK) monoclonal antibody (Thermo Scientific) at a dilution of 1:500 in TBST with 5% skim milk at 4 °C overnight. After washing with TBST, membranes were incubated with a horseradish peroxidase-conjugated goat anti-mouse IgG secondary antibody (AbFrontier, Seoul, Republic of Korea) at a dilution of 1:2000 in TBST with 5% skim milk for 2 h. Chemiluminescent detection was performed using the SuperSignal West Femto (Thermo Scientific). Blotting images were captured using Fusion FX6 Edge (Vilber-Lourmat, Marne-la-Vallée, France).

### 4.7. Isolation of Lipid A and MALDI-TOF MS Analysis

*A. baumannii* strains were grown in LB overnight at 37 °C, and cultured bacteria were inoculated into a fresh LB with pH 7.0 or 5.5, where the initial OD_600_ was 0.02. Bacteria were grown under shaking conditions until OD_600_ reached 0.25. The bacterial cultures (100 mL) were centrifuged at 13,000 rpm for 10 min. LPS was extracted using the LPS extraction kit (iNtRON, Seongnam, Republic of Korea) according to the manufacturer’s instructions. Briefly, bacterial cells were lysed by the lysis buffer, and the samples were vortexed vigorously. After adding chloroform to cell lysates, the samples were vortexed vigorously and incubated for 5 min. The mixture was centrifugated at 13,000 rpm for 10 min at 4 °C, and the purification buffer was added to the supernatants. The mixtures were incubated for 30 min at −20 °C, centrifuged at 13,000 rpm for 15 min at 4 °C, and removed from the upper layer to obtain LPS pellets. LPS pellets were washed with 70% ethanol and centrifuged at 13,000 rpm for 3 min at 4 °C. The upper layer was discarded, and LPS pellets were dried using a speed vacuum. Lipid A was isolated from LPS by mild-acid hydrolysis [36]. LPS pellets were dissolved in 1% acetic acid and hydrolyzed by boiling at 100 °C for 2 h. The samples were incubated with chloroform and methanol, and the phases were separated by centrifugation at 8000 rpm for 5 min at 15 °C. The lower phase (chloroform layer) was extracted and dried in the air. The dried lipid A was solubilized in chloroform/methanol (4:1, *v/v*). α-cyano-4-hydroxycinnamic acid (10 mg/mL) in TA50 (50% acetonitrile/0.1% trifluoroacetic acid) solution was used as a matrix. MALDI plates were spotted with the sample and matrix solution at a 1:1 (*v/v*) ratio, and lipid A structural spectra were acquired using reflector mode on the Autoflex max MALDI-TOF/TOF MS (Bruker Daltonics, Bremen, Germany) [37].

### 4.8. RNA Isolation and qPCR

*A. baumannii* strains were grown in LB overnight at 37 °C and inoculated into a fresh LB with pH 7.0 or 5.5, where the initial OD_600_ was 0.02. Bacteria were grown under shaking conditions until the OD_600_ reached 0.25. Total RNA was isolated using the RNeasy Mini kit (Qiagen, Valencia, CA, USA) according to the manufacturer’s instructions. Complementary DNA was synthesized by reverse transcription of 2 μg total RNA as a template using a TOPscript™ cDNA Synthesis Kit (Enzynomics, Daejeon, Republic of Korea). Quantification of gene transcripts was performed by real-time PCR using TOPreal™ qPCR 2X PreMIX (SYBR Green with high ROX; Enzynomics) using the StepOnePlus Real-Time PCR System (Applied Biosystems, Foster City, CA, USA). Gene expression was normalized to 16S rRNA levels, and fold change was calculated by the ΔΔCt method [38]. The primers used in this study are listed in Appendix A. Gene expression assays were performed in three independent experiments.

### 4.9. Statistical Analysis

Data were analyzed using GraphPad Prism 5.0 (San Diego, CA, USA). Averages and standard deviations of the mean were calculated from at least three independent experiments. Data from different experimental groups were analyzed using one-way ANOVA with Dunnett’s post hoc analysis or Student’s *t*-test. *p* < 0.05 was considered statistically significant.

## Figures and Tables

**Figure 1 antibiotics-12-00813-f001:**
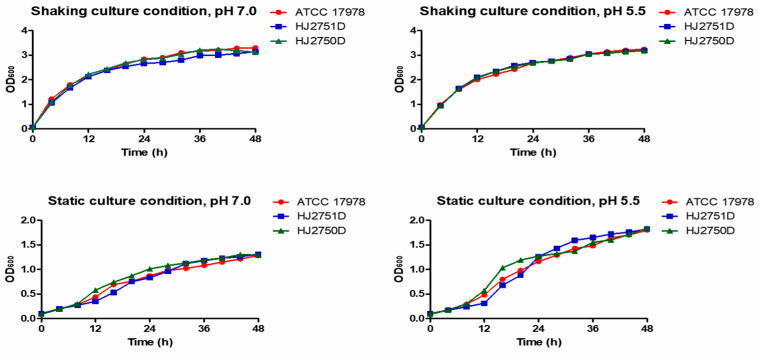
Growth of *A. baumannii* strains under different culture conditions. *A. baumannii* ATCC 17978, HJ2751D, and HJ2750D strains were grown in LB with the initial pH of 7.0 or 5.5 under shaking or static conditions. Bacterial growth was measured at the indicated time points. Data are representative of two experiments with similar results.

**Figure 2 antibiotics-12-00813-f002:**
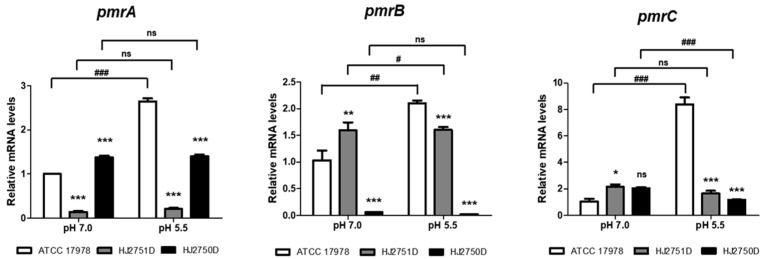
The expression of *pmrCAB* in *A. baumannii* strains cultured under different pH conditions. WT ATCC 17978, HJ2751D, and HJ2750D strains were grown in LB at pH 7.0 or 5.5 until OD_600_ reached 0.25. The expressions of *pmrA*, *pmrB*, and *pmrC* were determined by qPCR. The data are presented as mean ± SD of three independent experiments. * *p* < 0.05, ** *p* < 0.01, *** *p* < 0.001 compared to the WT strain. # *p* < 0.05, ## *p* < 0.01, ### *p* < 0.001 compared to bacteria grown at pH 7.0. ns, no statistically significant difference.

**Figure 3 antibiotics-12-00813-f003:**
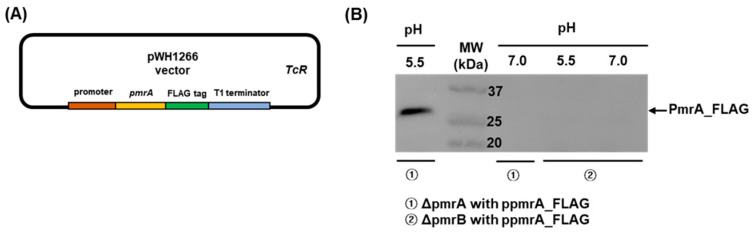
The production of PmrA protein in the Δ*pmrA* mutant strain carrying ppmrA_FLAG plasmids in response to acidic conditions. (**A**) Schematic diagram of the construction of ppmrA_FLAG plasmids. (**B**) Western blot analysis of bacterial extracts prepared from Δ*pmrA* or Δ*pmrB* mutant strains carrying ppmrA_FLAG plasmids. Bacteria were grown in LB at pH 7.0 or 5.5. Equal amounts of bacterial proteins were run in 10% SDS-PAGE gel and PmrA-FLAG proteins were detected using a FLAG Epitope Tag monoclonal antibody.

**Figure 4 antibiotics-12-00813-f004:**
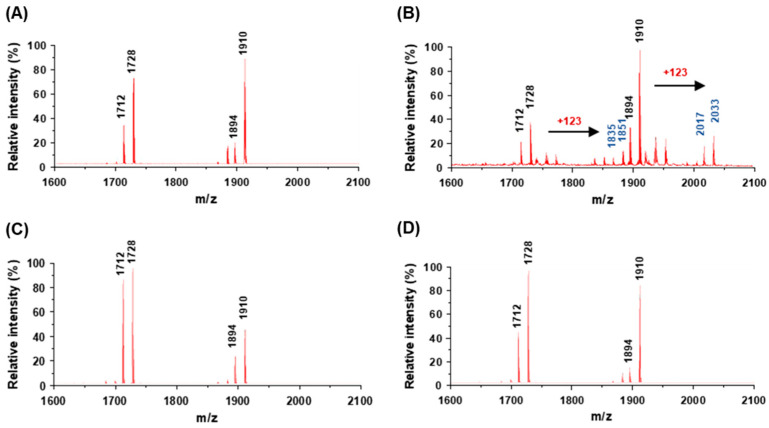
MALDI/TOF MS analysis of lipid A isolated from *A. baumannii* strains grown under different pH conditions. (**A**) *A. baumannii* ATCC 17978 strain grown at pH 7.0. (**B**) *A. baumannii* ATCC 17978 strain grown at pH 5.5. (**C**) Δ*pmrA* mutant (HJ2751D) strain grown at pH 5.5. (**D**) Δ*pmrB* mutant (HJ2750D) strain grown at pH 5.5. Lipid A isolated from *A. baumannii* ATCC 17978 grown at pH 7.0 showed the prototype lipid A with *m/z* 1712, 1728, 1894, and 1910 in the spectrum. *A. baumannii* ATCC 17978 grown at pH 5.5 showed four additional peaks of a shift of *m/z* +123 from four main peaks, corresponding to the addition of a single phosphoethanolamine residue.

**Table 1 antibiotics-12-00813-t001:** Effects of *pmrAB* on the colistin susceptibility of *A. baumannii* strains under different culture conditions.

Culture Conditions			Colistin MIC (μg/mL)			
			ATCC 17978	HJ2751D	HJ2750D	HJ2751C
CAMHB *	pH	pH 7.0	1	1	1	1
		pH 6.5	1	1	1	1
		pH 5.5	32	2	2	256
		pH 4.5	-	-	-	-
CAMHB pH 7.0	Fe^3+^	100 μM FeCl_3_	1	2	2	2
		500 μM FeCl_3_	4	4	4	4
		1 mM FeCl_3_	8	8	8	8
CAMHB pH 5.5	Fe^3+^	1 mM FeCl_3_	16	16	16	256

* CAMHB, cation-adjusted Mueller–Hinton broth.

**Table 2 antibiotics-12-00813-t002:** Bacterial strains and plasmids used in this study.

Bacteria/Plasmids	Relevant Characteristics *	Reference of Source
Bacterial strains		
*A. baumannii*		
ATCC 17978	Wild-type strain	ATCC
HJ2751D	∆*pmrA* of *A. baumannii* ATCC 17978	This study
HJ2750D	∆*pmrB* of *A. baumannii* ATCC 17978	This study
HJ2751C	*pmrA* rescued in ∆*pmrA* with pWH1266	This study
*E. coli*		
DH5α pir (SY327 λ pir)	*supE44* Δ*lacU169* (Φ80 *lacZ*ΔM15*) hsdR17 recA1 endA1 gyrA96 thi-1 relA1 λpir* (phage lysogen); plasmid replication	Laboratory collection
sm10λ pir	*thi thr leu tonA lacY supE recA::RP4-2-Tc::Mu Km λpir* π-requiring plasmids; conjugal donor	Laboratory collection
Plasmids		
pOH4	pHKD01 with the *ompA* coding region of *A. baumannii* ATCC 17978 under the control of its native promoter with *nptI*; Km^R^	Laboratory collection
pDM4	Suicide vector, *ori* R6K; Cm^R^; sacB	GenBank accession no. KC795686
pWH1266	Shuttle-vector with *Acinetobacter* and *E. coli* origin used for cloning vehicle; amp^R^; tet^R^	Laboratory collection
pHJ2751D	pDM4 with ∆*pmrA*::*nptI*; Cm^R^, Km^R^	This study
pHJ2750D	pDM4 with ∆*pmrB*::*nptI*; Cm^R^, Km^R^	This study
pHJ2751C	pWH1266 carrying *pmrA* with the *ompA* promoter and T1 terminator; Tet^R^	This study
ppmrA_FLAG	pWH1266 carrying the FLGA-tagged *pmrA* coding region under the control of its native promoter with t1 terminator; Tet^R^	This study

* Cm^R^, chloramphenicol resistant; Km^R^, kanamycin resistant; Amp^R^, ampicillin resistant; Tet^R^, tetracycline resistant.

## Data Availability

The authors confirm that the data supporting the findings of this study are available within the article and its Appendix A.

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
