# Peer review of "Acinetobacter baumannii under Acidic Conditions Induces Colistin Resistance through PmrAB Activation and Lipid A Modification"

_antibiotics, 2023, doi:10.3390/antibiotics12050813_

Round 1

Reviewer 1 Report

The provided manuscript describes colistin resistance induction in A. baumannii under acidic conditions explained as a cause of pmrCAB operon activation and subsequent lipid A modification.

In general, the manuscript is well written, the information presented is well organized. The findings would have further scientific implications and it would be interesting to the readers of Antibiotics. I only have some comments which I would like the authors to address:

Phage 75: Suggest to remove this “Instead of its promoter”- or write it was replaced.

Page 84: Suggest to write “of two different initial pH values of 7.0 and 5.5 separately “instead of “where the initial pH of the culture medium was 7.0 or 5.5.” and was pH 7.0 selected as a control to acidic one? If so, indicate it.

Page 91: better to write: “To determine the effects of pH on”

Page 93: suggest write “of different pH values” rather “with different pH levels”

Page 108-109: suggest to give full title to each graph of figure 1: “shaking culture condition, pH..”

Page 248: suggest to write growth media rather conditions “bacterial strains, plasmids, and growth conditions”

Page 250: Refer provider of “LB or LB “add if it was prepared at lab.

Add here “CAMHB and colistin” as a materials and refer all.

Page 251: Refer provider of “HCl or NaOH “add if it was prepared at lab.

Page 251-252: remove this sentence « E. coli DH5α pir (Sy327 λ pir) and E. coli sm10λ pir were used to construct A. baumannii mutant strains«. as strains are listed in the table 2; and mutant construction in the section of methods.

Page 252-253:

This sentence - “To maintain plasmids in bacteria and select the mutant or complementary colonies”- should go to the section of methods

Refer the antibiotics used: “kanamycin (50 μg/ml), chloramphenicol (20 μg/ml), or tetracycline (10 μg/ml) were added to the growth medium.”

Page 280: Does this” MICs were defined as the lowest colistin concentration that prevents the visible growth of bacteria” mean that results were evaluated visually? Then describe wow was evaluated different levels of clearance.

Otherwise refer the machine used for.

Page 285-86: here it is better two write sampling time intervals “Bacteria were sampled at the indicated time points” or refer the figure or table provided above.

Page 292: again, it is better two write pH values “fresh LB with the indicated pH” or refer the figure or table provided above.

Page 290-309: Give reference of the method - 4.6. Western blot analysis

Page 317: in the methods the values of used pH should be written – “LB with the indicated pH”.

Page 315-337: Give reference of the method “4.7. Isolation of lipid A and MALDI-TOF MS analysis

Page 341: again, it is better two write pH values “LB with the indicated pH”

Page 339-351: Give reference of the method - «4.8. RNA isolation and qPCR« 

Minor editing of English language required

Author Response

The provided manuscript describes colistin resistance induction in A. baumannii under acidic conditions explained as a cause of pmrCAB operon activation and subsequent lipid A modification. In general, the manuscript is well written, the information presented is well organized. The findings would have further scientific implications and it would be interesting to the readers of Antibiotics. I only have some comments which I would like the authors to address:

Page 75: Suggest to remove this “Instead of its promoter”- or write it was replaced. 

As your suggestion, we deleted it.

Page 84: Suggest to write “of two different initial pH values of 7.0 and 5.5 separately “instead of “where the initial pH of the culture medium was 7.0 or 5.5.” and was pH 7.0 selected as a control to acidic one? If so, indicate it.

As your suggestion, we changed it as follow: Bacterial strains were grown in lysogeny broth (LB), where the initial pH of the culture medium was 7.0 and 5.5 separately. pH 7.0 was selected as a control to acidic pH.

Page 91: better to write: “To determine the effect of pH on”

As your suggestion, we changed it as follow: To determine the effect of pH on.

Page 93: suggest write “of different pH values” rather “with different pH levels”

As your suggestion, we changed it.

Page 108-109: suggest to give full title to each graph of figure 1: “shaking culture condition, pH.”

As your suggestion, we changed the subtitle of each figure and Figure 1 was replaced with new figure.

Page 248: suggest to write growth media rather conditions “bacterial strains, plasmids, and growth conditions”

As your suggestion, we changed it.

Page 250: Refer provider of “LB or LB “add if it was prepared at lab. Add here “CAMHB and colistin” as a materials and refer all.

As your suggestion, we added the provider of LB.

Page 251: Refer provider of “HCl or NaOH “add if it was prepared at lab.

As your suggestion, we added the provider of HCL and NaOH.

Page 251-252: remove this sentence « E. coli DH5α pir (Sy327 λ pir) and E. coli sm10λ pir were used to construct A. baumannii mutant strains«. as strains are listed in the table 2; and mutant construction in the section of methods.

As your suggestion, we deleted it.

Page 252-253:

This sentence - “To maintain plasmids in bacteria and select the mutant or complementary colonies”- should go to the section of methods

Refer the antibiotics used: “kanamycin (50 μg/ml), chloramphenicol (20 μg/ml), or tetracycline (10 μg/ml) were added to the growth medium.”

As your suggestion, we changed it.

Page 280: Does this” MICs were defined as the lowest colistin concentration that prevents the visible growth of bacteria” mean that results were evaluated visually? Then describe wow was evaluated different levels of clearance.

Otherwise refer the machine used for.

As described in the text, colistin MICs were determined using the naked eyes. The growth of bacterial in the wells containing colistin was compared to the control well without bacteria. Complete inhibition of bacterial growth was defined to MIC.

Page 285-86: here it is better two write sampling time intervals “Bacteria were sampled at the indicated time points” or refer the figure or table provided above.

As your suggestion, we changed it as follows: Bacteria were sampled at 4 h intervals,

Page 292: again, it is better two write pH values “fresh LB with the indicated pH” or refer the figure or table provided above.

As your suggestion, we changed it as follows: Cultured bacteria were inoculated into a fresh LB with pH 7.0 or 5.5,

Page 290-309: Give reference of the method - 4.6. Western blot analysis

As your suggestion, we added reference 35.

Page 317: in the methods the values of used pH should be written – “LB with the indicated pH”.

As your suggestion, we added the pH values.

Page 315-337: Give reference of the method “4.7. Isolation of lipid A and MALDI-TOF MS analysis

As your suggestion, we added reference 37.

Page 341: again, it is better two write pH values “LB with the indicated pH”

As your suggestion, we added the pH values.

Page 339-351: Give reference of the method - 4.8. RNA isolation and qPCR 

As your suggestion, we added reference 38.

Reviewer 2 Report

The authors demonstrated that A. baumannii under acidic conditions induces colistin resistance via activation of pmrCAB operon and subsequent modification by establishing pmrA and pmrB deletion mutants. These results are clearly and acceptable data. It is expected that the mechanisms of colistin resistance in A. baumannii will be understood.

Author Response

The authors demonstrated that A. baumannii under acidic conditions induces colistin resistance via activation of pmrCAB operon and subsequent modification by establishing pmrA and pmrB deletion mutants. These results are clearly and acceptable data. It is expected that the mechanisms of colistin resistance in A. baumannii will be understood.

Many thanks for your review of our manuscript. The manuscript will extend our knowledge of colistin resistance associated with two-component system PmrAB in A. baumannii.

Reviewer 3 Report

In my opinion, it is an excellent piece of research that reveals the mechanisms that may be associated with the antimicrobial resistance of A. baumannii to colistin.

With excellently designed and performed studies, it describes the effect of acidic pH and the presence of ferric chloride on the structure and integrity of LPS and its relationship with resistance to Colistin.

I believe it would be pertinent to emphasize the microenvironmental conditions that can be generated during infection, which could be associated with an acidic pH, high irons, and low magnesium concentrations.

Author Response

In my opinion, it is an excellent piece of research that reveals the mechanisms that may be associated with the antimicrobial resistance of A. baumannii to colistin. With excellently designed and performed studies, it describes the effect of acidic pH and the presence of ferric chloride on the structure and integrity of LPS and its relationship with resistance to Colistin.

I believe it would be pertinent to emphasize the microenvironmental conditions that can be generated during infection, which could be associated with an acidic pH, high irons, and low magnesium concentrations.

Many thanks for your review of our manuscript. The manuscript will extend our knowledge of colistin resistance associated with two-component system PmrAB in A. baumannii.